# Prediction Model and Mechanism for Drying Shrinkage of High-Strength Lightweight Concrete with Graphene Oxide

**DOI:** 10.3390/nano13081405

**Published:** 2023-04-19

**Authors:** Xiaojiang Hong, Jin Chai Lee, Jing Lin Ng, Muyideen Abdulkareem, Zeety Md Yusof, Qiansha Li, Qian He

**Affiliations:** 1Department of Civil Engineering, Faculty of Civil and Hydraulic Engineering, Xichang University, Xichang 615013, China; hxjxcxy@126.com (X.H.);; 2Department of Civil Engineering, Faculty of Engineering, Technology and Built Environment, UCSI University, Kuala Lumpur 56000, Malaysia; 3Department of Civil Engineering, Faculty of Civil Engineering and Built Environment, Universiti Tun Hussein Onn Malaysia, Johor 86400, Malaysia

**Keywords:** drying shrinkage, modified prediction model, pore structure, microstructure, high-strength lightweight concrete, shale ceramsite, graphene oxide

## Abstract

The excellent performance of graphene oxide (GO) in terms of mechanical properties and durability has stimulated its application potential in high-strength lightweight concrete (HSLWC). However, more attention needs to be paid to the long-term drying shrinkage of HSLWC. This work aims to investigate the compressive strength and drying shrinkage behavior of HSLWC incorporating low GO content (0.00–0.05%), focusing on the prediction and mechanism of drying shrinkage. Results indicate the following: (1) GO can acceptably reduce slump and significantly increase specific strength by 18.6%. (2) Drying shrinkage increased by 8.6% with the addition of GO. A modified ACI209 model with a GO content factor was demonstrated to have high accuracy based on the comparison of typical prediction models. (3) GO not only refines the pores but also forms flower-like crystals, which results in the increased drying shrinkage of HSLWC. These findings provide support for the prevention of cracking in HSLWC.

## 1. Introduction

Lightweight aggregate concrete (LWAC) is applied extensively in civil construction engineering and conforms to the trend of environmental protection and sustainable development. LWAC has been used as a non-structural material for load reduction, noise reduction, and thermal insulation for decades [1,2]. With the promotion of advanced concrete technology, LWAC, as a potential structural material, is urgently upgraded to high-strength lightweight concrete (HSLWC) with better mechanical properties and excellent durability, thus realizing the fabrication of large-scale structures such as super high-rise buildings, long-span bridges, deep-sea structures, and prefabricated buildings [3,4]. Nevertheless, raising compressive strength to 55 MPa is a challenging obstacle for HSLWC because of the porous structure and low-strength characteristics of the aggregates [5]. Several studies on using plastic waste and recycled aggregates to manufacture concrete have been extracted into practically valuable results [6,7]. Lee et al. reported that HSLWC with a 28-day compressive strength of 55 MPa and an oven dry density of 1905 kg/m^3^ was manufactured with oil-palm-boiler clinker (OPBC) as the coarse aggregate [8]. Sajedi et al. investigated this form of concrete and found that HSLWC with 28-day compressive strength in the range of 34–79 MPa and oven dry density in the range of 1610–1965 kg/m^3^ was produced when using expanded aggregate clay (LECA) under various substitution conditions [9]. Kockal et al. found that HSLWC with a 28-day compressive strength of 56 MPa and an oven density of 1943 kg/m^3^ was prepared using fly ash (FA) [10].

In China, the vigorous development of engineering construction has not only caused the depletion of natural river sand, but also accumulated a large amount of shale spoil. Through a high-temperature calcination process, shale ceramsite (SC) and shale pottery sand (SPS) can be prepared from shale spoil on an industrial scale [11]. SC is widely considered as a good coarse aggregate for structural HSLWC due to its low density, high strength, good seismic performance, and high porosity [12]. Based on these characteristics, the prewetted SC exhibits a unique internal curing system, which fully ensures the hydration of the early-age cement paste and effectively relieves autogenous shrinkage [13]. In addition, the infiltration of hydration products into the pores results in a denser and more uniform bond in the interfacial zone between SC and cement compared to normal-weight concrete (NWC) [14]. In general, concrete containing SC is not as sound in terms of mechanical properties as NWC. Therefore, it is necessary to use additives to further improve performance and expand the application space.

The advantage of adjusting or modifying cement-based materials at the nanoscale brings positive prospects for the widespread application of nanomaterials in the concrete industry. Graphene oxide (GO) is one of the most typical nanomaterials with a unique two-dimensional structure. Currently, GO can enhance the mechanical properties of cement mortar or ultra-high performance concrete (UHPC) due to greater mechanical characteristics, better adhesion, and superior surface area [15]. Lv et al. confirmed that GO can effectively regulate the morphology of hydration crystals, thus increasing the compressive strength and splitting tensile strength of cement mortar by 60% and 96%, respectively [16]. Wu et al. claimed that the addition of GO in the range of 0.02–0.08% can improve the compressive strength of NWC (water to cement ratio of 0.5) by 12.8–34.1% [17]. They also found that the optimal amount (0.02%) of GO can maximize the compressive strength and flexural strength of UHPC by 28.7% and 25.3%, respectively [18]. Chu et al. obtained a similar finding, but the difference was that the optimal GO addition required for UHPC containing recycled sand was estimated to be 0.05% [19]. These conclusions also show that the optimal amount of GO is related to the raw materials of the mixture. Furthermore, GO also provides a positive contribution to the durability of NWC and UHPC. Xu et al. observed that the compressive strength of a mixture containing 0.03% GO was 34.8% higher than that without GO after 200 salt-freezing cycles [20]. Yu et al. ascertained that when the optimal amount of GO added is 0.06%, the chloride ion migration coefficient of UHPC can be reduced by 4.3% [21]. Despite the advantages of concrete containing GO compared to NWC, rapid improvement in mechanical properties and durability may accelerate the drying shrinkage that can shorten the serviceability of structures [22].

Drying shrinkage, as the time-dependent deformation in the unloaded state, has to be considered in practical applications for HSLWC. The pore structure of concrete is very essential for its drying shrinkage as it determines its porosity and ability to lose water. Drying shrinkage is fundamentally restricted by the volume change caused by water loss in the transition pores and the gel pores. The loss of water in capillary pores causes shrinkage stress [23]. There are many factors that affect the drying shrinkage of concrete, such as the type and content of aggregate, the water cement ratio, the pore structure, and the external environment. A previous study found that LWAC made with OPBC had higher drying shrinkage strain than NWC, mainly owing to the properties and volume of the lightweight aggregates [24]. Newman reported that the drying shrinkage of HSLWC made with dense fine aggregate was almost equal to that of NWC, which can be estimated as about 350 micro-strains in the absence of measurement [25]. For common structures, the shrinkage of HSLWC can be considered to be 1.2–2 times that of NWC [26]. However, Shafigh et al. reported that HSLWC presented a lower shrinkage than that of NWC due to the supply of internal curing water stored in the pores [27]. Furthermore, precise and reliable prediction of drying shrinkage deformation development is difficult. Various models from different codes and standards have been developed to predict drying shrinkage under specific conditions. Their equations take into account different internal factors and external factors. Relevant studies showed that these models obtain varying degrees of accuracy compared with experimental results [28,29]. In addition, some researchers have innovatively modified the typical drying shrinkage models according to other additives and influence factors [30,31].

Although some progress has been made in using GO to enhance mechanical properties, research on the effect of GO on drying shrinkage is still in the exploratory stage. Moreover, Chen and Xu are among the very few who have established a prediction model for drying shrinkage of NWC involving the modification of GO [32]. At present, the effect of GO on the drying shrinkage of HSLWC containing SC has been rarely explored. The aims of this study were to investigate the effect of different GO content on the drying shrinkage behavior of HSLWC and to analyze its mechanism in terms of pore structure and microstructure. This study also tried to establish a modified model considering the GO content factor on the basis of comparing the accuracy of typical prediction models.

## 2. Materials and Methods

### 2.1. Materials

The binder used in the test was grade 42.5 R ordinary Portland cement, which was manufactured by Xichang Aerospace Co., Ltd. (Xichang, China). The apparent density and 28-day compressive strength were 3080 kg/m^3^ and 47.4 MPa, respectively. In order to meet the requirement of being lightweight, SPS and SC were selected as the fine aggregate and coarse aggregate, respectively, which were provided by Hubei Huiteng Aggregate Co., Ltd. (Yichang, China). The physical characteristic parameters are listed in Table 1. Specifically, Figure 1 presents the appearance and microstructure of SC. It was obvious that SC had a rough, angular exterior and a porous honeycomb interior. SPS with a fineness modulus of 2.96 conformed to the standard of medium sand in JGJ52-2016, and the cumulative particle size distributions of SPS are shown in Figure 2. The mineral additive used in the study was Class I fly ash (FA) with a specific surface area of 420 m^2^/kg, which was produced by a local company. A superplasticizer (SP) with a water reduction rate of 12% was obtained from Weihe Co., Ltd. (Xian, China).

The GO used in this study was a high-purity, brownish powder obtained from Suzhou Tanfeng Technology Graphene Technology Co., Ltd. (Suzhou, China), and its properties are listed in Table 2. Despite the presence of hydrophilic functional groups, GO powders have different critical points of dissolution in different solvents [33]. A previous study proved that low doses of GO powders can be uniformly dispersed in water by using SP as an active agent [34], and the GO dispersion is shown in Figure 3a. As shown in Figure 3b, GO was a nanoscale sheet with a typical fine and dense wrinkle morphology when measured by SEM. This special morphology can contribute to filling or refining the pores of the structure due to its excellent internal strength and large surface area.

### 2.2. Mix Proportions

A control mix proportion (HS0) with a high-strength grade of 60 was designed according to standard GB/T 31387-2015. Low GO content can significantly change the performance of concrete, but high GO content can attenuate the strengthening effect. Herein, the other three groups (HS1, HS3, and HS5) were experimentally designed. Correspondingly, the additive content of GO was 0.01%, 0.03%, and 0.05% (by weight of cement), respectively. To verify the effectiveness of GO in terms of the drying shrinkage of HSLWC, all groups had the same other raw materials and a water to binder ratio of 0.31, as shown in Table 3.

### 2.3. Test Methods

Prior to casting, two preparation steps needed to be completed: (1) SC was prewetted for 24 h and then subjected to saturated surface drying conditions during casting. (2) GO powder and SP were mixed in water and dispersed for 30 min by an ultrasonic dispersion device to form a GO dispersion. It was noted that the GO dispersion should not be left for too long so as to avoid aggregation. The detailed casting process is summarized as follows: (1) SC and SPS were mixed in the rotary drum mixer and stirred for 2 min. (2) Cement and FA was added, and stirring was continued for 2 min. (3) Approximately 70% of the GO dispersion was added to the mixture obtained in the previous step, with stirring for another 3 min. (4) The remaining GO dispersion was added, followed by stirring for a further 2 min.

Before molding, the fresh concrete was subjected to slump tests according to the procedure outlined in GB/T 50080-2016. The 100 mm cube specimens were cast for the testing of 28-day compressive strength according to the procedure outlined in GB/T 50081-2002. The drying shrinkage test was performed according to procedure outlined in GB/T 50082-2009. A vertical shrinkage measuring device (NELD-ES700) was used to measure the drying shrinkage of HSLWC, which required specimens with dimensions of 515 mm × 100 mm ×100 mm. When age reached 3 days under standard curing, the specimens were moved into another room (temperature: (20 ± 2) °C; relative humidity: (60 ± 5)%), with the initial length subsequently recorded. In order to analyze the drying shrinkage trend of HSLWC, the readings of all the dial gauges were recorded at 1d, 3d, 7d, 14d, 28d, 45d, 60d, 90d, 120d, 150d, 180d, and 360d. The drying shrinkage value was calculated according to Equation (1).
(1)$$\varepsilon_{st}=\frac{L_{0}-L_{t}}{L_{b}}$$
where *ε_st_* is the measured drying shrinkage value at the age of t (μm/mm), t is the test age of the specimen (d), *L*_0_ is the initial value of the concrete specimen’s length (mm), *L_t_* is the measured value of the concrete specimen’s length at the test age of t (mm), and *L_b_* is the gauge length of the concrete (mm).

The mechanism by which GO affects the drying shrinkage of HSLWC needed to be explored. On the one hand, paste flakes selected from crushed specimens of each group with an age of 28 days were examined for crystal morphology using scanning electron microscopy (SEM: Thermo Scientific Apreo 2C). On the other hand, typical blocks made from the same specimens that participated in the SEM test were observed for their porosity and pore size distribution by mercury intrusion porosimetry (MIP: MicroActive AutoPore V 9600). The detailed mixing procedures and experimental items are shown in Figure 4.

## 3. Results and Discussion

### 3.1. Workability and Compressive Strength

Table 4 lists the slump results of different mixtures. It can be observed that the slumps of the specimens with GO were significantly lower than those without GO, presenting a decreasing trend with the increase in GO content. Compared with the control mixture (HS0), the slump of the mixture was reduced by 25.6% when GO content was 0.05%. This result might be attributed to good adhesion and the large specific surface area of GO. The adverse impact of GO on slump has also been confirmed in UHPC [18]. Mehta et al. claimed that LWAC obtained the same acceptable workability as NWC when slump value exceeded the range of 50–75 mm [35]. Obviously, the slump of 84–113 mm in this study met this requirement.

It can also be seen from Table 4 that with the increase in GO content, the oven dry densities of the mixtures were almost constant, while 28-day compressive strength increased significantly. Compared with HS0, the 28-day compressive strengths of HS1, HS3, and HS5 increased by 2.9%, 9.6%, and 20.1%, respectively. Moreover, 28-day compressive strength was positively related to GO content, as presented in Figure 5. The results indicate that adding lower GO content can yield better benefits in terms of improved mechanical properties. Metha et al. classified concrete with oven dry density less than 1850 kg/m^3^ and 28-day compressive strength in the range of 34–79 MPa as HSLWC [35]. Accordingly, the concrete in this study can be categorized as HSLWC. Specific strength is an important parameter to evaluate the structural performance of HSLWC, which is the ratio of compressive strength to weight. The specific strength of the control mixture (HS0) was 36.5 kN·m/kg, while that of NWC was 18.6 kN·m/kg according to Moravia et al. [36]. Related studies reported that the specific strengths of HSLWC produced with OPBC and expanded shale as coarse aggregate were 30.9 and 36.3 kN·m/kg [37,38]. Additionally, the specific strength of the mixture with GO increased by at most 18.6%, which was mainly due to the improvement in compressive strength promoted by the addition of GO.

### 3.2. Drying Shrinkage

#### 3.2.1. Experimental Result

Figure 6 clearly displays the development of the drying shrinkage of specimens with different GO content over the course of 360 days. For the control mixture (HS0), drying shrinkage behavior was divided into three stages. (1) Rapid rise stage: in the first 28 days, drying shrinkage increased sharply, which was mainly due to the reduction in volume and free water caused by the paste hydration reaction. (2) Slow rise stage: over the next 122 days, the increase in drying shrinkage gradually slowed down due to the combination of continuous hydration and a constant supply of internal curing water from the prewetted SC. (3) Stable stage: In the last 180 days, the drying shrinkage value tended to be constant and was finally measured at 465 μm/m. This was close to the drying shrinkage value of LWAC produced by SC and sand reported by Gong et al. [39], though smaller than the value of HSLWC produced by OPBC reported by Aslam et al. [26]. Aslam also found that replacing oil palm shell (OPS) with different proportions of OPBC reduced drying shrinkage, which was caused by a reduction in the water absorption of aggregates. In contrast, drying shrinkage was smaller because (1) the prewetting process of the aggregates provided internal curing water to the microporous structure system of the paste, thus inhibiting the later drying shrinkage [40], and (2) after the same level of prewetted treatment, the higher the water absorption of aggregate, the more free water lost in the hydration reaction process. The water absorption of SC was lower than that of OPBC.

In addition, it should be mentioned that the drying shrinkage values measured in the HSLWC with GO were slightly higher than those without GO at each age, presenting a similar increasing trend. Obviously, the drying shrinkage of the specimens containing GO (HS1, HS3, HS5) had a sharper growth in the first stage. The drying shrinkage values of HS1, HS3, and HS5 at 360 days were 471, 485, and 505 μm/m, respectively. Compared with HS0, drying shrinkage increased by 1.2%, 4.3%, and 8.6%, respectively, indicating that dry shrinkage increased with the increase in GO content. The drying shrinkage of NWC was reported to be in the range of 200–800 μm/m at the age of 180 days [41], while that of HSLWC containing sintered fly ash could reach 1000 μm/m at the age of 360 days [42]. The addition of GO increased drying shrinkage by 1.2–8.6% in this study. However, this range was 1.33–6.72% when GO was mixed in UHPC [21]. It can be concluded that the HSLWC in this study still had an acceptable range of drying shrinkage values, even if the addition of GO caused adverse effects to some extent.

The increase in drying shrinkage caused by GO might be due to the following reasons. (1) GO accelerated the hydration reaction during the early stage to develop a dense microstructure but caused a rapid reduction in structure volume. (2) The development of drying shrinkage was essentially a process in which the free water in the hydration reaction was removed from gel pores and transition pores [43]. Consequently, the greater the amount of such pores, the more serious the drying shrinkage. GO, with a nanoscale folding structure, could refine pores to form many gel pores and transition pores in the hydration reaction. (3) GO absorbed some free water during the rise stages because of its special folding structure. The adsorbed water was then released to alleviate drying shrinkage during the stable stage.

#### 3.2.2. Numerical Comparison of Typical Models

When used as a structural material, HSLWC might cause fatal damage to structures due to excessive shrinkage. Therefore, it is necessary to master the long-term development trend of drying shrinkage. Various complicated factors were not conducive to deriving a common and accurate drying shrinkage function from the mechanism. Instead, many empirical prediction models from different codes and standards were developed, which depend on various factors to adapt specific conditions. At present, these typical models have been verified by a large amount of experimental data, including data from the China Academy of Building Research (CABR), CEB-FIP, ACI 209, B3, GL2000, and SAKATA, and their details are illustrated in Table 5.

Drying shrinkage development needs to be accurately predicted, and if not considered and restricted, can threaten the durability and safety of a structure throughout the whole life cycle. Due to the deterministic characteristics of time-dependent deformation, using hyperbolic functions to simulate drying shrinkage behavior has been widely explored and optimized. Taking into account specific influencing factors, the above-mentioned models have verified applicability and accuracy under various conditions through a large amount of experimental data. Specifically, different models are employed to simulate the same set of experimental data at different levels of accuracy. Only if these factors are set sufficiently is prediction accuracy expected to be better. The factors of the above models are summarized in Table 6.

Figure 7 displays a comparison of the experimental and predicted results for the control mixture (HS0) under the premise that all the factors suggested by the above model are set uniformly. In addition, Figure 8 also displays the average percentage errors of different prediction models in stage 1 (1–28 days) and stage 2 (45–360 days).

As shown in Figure 7a, the CABR model seriously underestimated the measured value but presented a trend towards error reduction. On the one hand, the CABR model involved fewer factors, especially internal factors. On the other hand, the setting of the compressive strength factor only reached grade 30 of LWAC, with higher grades not elaborated upon. Therefore, the CABR model failed in the application of this study.

As shown in Figure 7b, the CEB-FIP model obtained the worst approximation from stage 1 to stage 2 and significantly overestimated the measured values. The CEB-FIP model has been proven to have high accuracy for NWC with a compressive strength lower than 60 MPa. The characteristics of HSLWC, such as the use of a lightweight aggregate, high strength, and low water–cement ratio, cannot be accurately quantified. In addition, the CEB-FIP model used a hyperbolic power function for simulation, which might lead to a sharper rise.

As shown in Figure 7c, the ACI 209 model was found to be the closest approximation throughout the test period. The average error percentage was generally less than 5%, reaching a minimum at the final shrinkage stage. This might be because both internal and external factors were justifiably included in the model.

As shown in Figure 7d, the B3 model deviated from the measured value in stage 1 and then gradually approached the measured value in stage 2. In general, the B3 model provided a better approximation of the measured values than the CABR model and CEB-FIP model in this study. The HSLWC in this study satisfied the constraints of the B3 model: water–cement ratio in the range of 0.30–0.85, cement content in the range of 160–720 kg/m^3^, and 28-day compressive strength in the range of 18–70 MPa.

As shown in Figure 7e, the GL2000 model had a similar approximation of measured values as the B3 model, which might be caused by the same factors being selected. Obviously, the average error percentage of the GL2000 model was lower than that of the B3 model, especially in stage 1.

As shown in Figure 7f, the SAKATA model moderately underestimated the measured values throughout the test period. The predicted values of the SAKATA model deviated from the measured values at the beginning, and afterwards the trend became more obvious as time progressed.

Based on the above analysis, these models revealed different levels of prediction accuracy. Firstly, the ACI 209 model has been proven to be the most accurate prediction of the measured value. Secondly, the SAKATA model showed excellent accurate prediction throughout the test period. Both the B3 model and the GL2000 model had reasonably accurate predictions, especially in stage 2. Finally, the CABR model and CEB-FIP model showed weakly accurate predictions.

#### 3.2.3. Model Modification

The ACI 209 model was confirmed in the previous section to best match the measured drying shrinkage of HSLWC without GO. The ACI 209 model, established by the American Concrete Institute (ACI), is one of the most popular models due to simple calculation and high accuracy. The formula of this model is derived from a drying shrinkage value of concrete under standard conditions that is modified by various factors. The applicable objects were mainly NWC and LWAC. The drying shrinkage value of the model was calculated by Equations (2) and (3).
(2)$$\varepsilon_{sh(t)}=\frac{t}{35+t}\times \varepsilon_{sh,\infty },$$
(3)$$\varepsilon_{sh,\infty }=780\times \gamma_{cp}\times \gamma_{a}\times \gamma_{c}\times \gamma_{h}\times \gamma_{\lambda }\times \gamma_{s}\times \gamma_{\varphi },$$
where $\varepsilon_{sh(t)}$ is the calculated drying shrinkage value at the age of t (μm/mm), $\varepsilon_{sh,\infty }$ is the ultimate drying shrinkage value (μm/mm), $\gamma_{cp}$ is the maintenance method factor, $\gamma_{a}$ is the air content factor, $\gamma_{c}$ is the cement content factor, $\gamma_{h}$ is the component section size factor, $\gamma_{\lambda }$ is the relative humidity factor, $\gamma_{s}$ is the slump factor, and $\gamma_{\varphi }$ is the fine aggregate content factor.

Figure 9 presents a comparison of the ACI 209 model and the experimental shrinkage values of HSLWC with different GO content. It was interesting to note that the predicted value of the ACI model decreased with increasing GO content. However, this prediction was contrary to the actual measurement. The real reason for this phenomenon was that all mixtures had the same factors, except for the slump factor ($\gamma_{s}$). In fact, the increase in GO content ($p_{GO}$) was negatively related to slump (s). According to Equation (4), the slump factor ($\gamma_{s}$) was positively related with the slump (s). Therefore, the predicted value was theoretically reduced.
(4)$$\gamma_{s}=0.89+0.00161\times s,$$

In order to further improve prediction accuracy, some studies have attempted to modify the ACI 209 model. Ou et al. used the power function to optimize the hyperbolic function for the prediction of high drying shrinkage in alkali-activated slag concrete [50]. Mushtaq et al. added a linear equation to the original model for the prediction of drying shrinkage in concrete containing waste foundry sand [51]. The increase in GO content inevitably led to a deterioration in prediction accuracy, so it was necessary to establish a more precise model containing the GO content factor. A GO content factor inspired by the slump factor was developed to modify the ACI 209 model.

According to Equation (3), $\varepsilon_{sh,\infty }$ involved the product of seven factors. Each factor was normalized to fairly reflect its degree of influence on drying shrinkage. Based on the experimental fact that drying shrinkage was positively related with GO content, this study attempted to introduce the GO content factor to correct the deviations in Figure 9. First, the ratio of the measured value to the predicted value was calculated for each test age. Subsequently, to reflect generality, the GO content factor ($\gamma_{GO}$) was defined as the average of the ratios of all test ages for each GO content value ($p_{GO}$). Finally, as shown in Figure 10, the fitting relationship between the GO content factor and GO content was found to have a good correlation, with a correlation coefficient of 0.98. The fitting equation is shown in Equation (5).
(5)$$\gamma_{GO}=1+240\times p_{GO},$$
where $\gamma_{GO}$ is the GO content factor and $p_{GO}$ is GO content.

GO not only accelerated the hydration reaction but also reduced volume, thus increasing compressive strength and drying shrinkage. The GO content factor essentially compensates for the modification of compressive strength in the ACI 209 model. This fitting was reasonable due to the fact that the relationship between compressive strength and GO content was proven to be linear in Figure 5. The modified ACI 209 model with the GO content factor is shown in Equations (6) and (7).
(6)$$\varepsilon_{sh(t)}=\frac{t}{35+t}\times \varepsilon_{sh,\infty }^{\prime },$$
(7)$$\varepsilon_{sh,\infty }^{\prime }=780\times \gamma_{cp}\times \gamma_{a}\times \gamma_{c}\times \gamma_{h}\times \gamma_{\lambda }\times \gamma_{s}\times \gamma_{\varphi }{\times \gamma }_{GO}.$$

Figure 11 presents a comparison of the modified ACI 209 model and experimental shrinkage values of HSLWC with different GO content. It was obvious that the predicted values of the modified ACI 209 model showed a reasonable trend of increasing with the increase in GO. In addition, the predicted value of the modified ACI 209 model was closer to the measured values. Therefore, the introduction of the GO content factor was reasonable. Figure 12 presents the relative errors of prediction with different GO content at each test age. It can be found that the relative errors of the entire prediction behavior were small, showing a trend of attenuation, ultimately being less than 5%. The average relative errors of HS0, HS1, HS3, and HS5 were 2.6%, 1.8%, 3.4%, and 3.3%, respectively, indicating that the modified ACI 209 model has high accuracy.

### 3.3. Pore Structure

According to the findings of Meng et al. [52], the pores of cement-based materials measured by MIP can be classified into the following types: gel pores (<10 nm), transition pores (10–100 nm), capillary pores (100~1000 nm), and macropores (>1000 nm). The total porosities of HS0, HS1, HS3, and HS5 were 0.1316, 0.1257, 0.1057, and 0.0829 mL/g, respectively. It can be seen that the total porosities of the specimens incorporating GO were significantly lower than those without GO, indicating that adding a low amount of GO content in HSLWC can effectively reduce total porosity. Figure 13 displays the percentages of pores of different pore size for each mixture. The changes in the pore structure of HSLWC caused by increasing GO content included (1) a decrease in macropores and capillary pores and (2) a slight increase in transition pores and gel pores.

Figure 14 displays the pore size distributions of individual mixtures. It can be clearly seen that all mixtures had similar differential pore size distributions accompanied by two typical peaks. The pore sizes of the two peaks appeared in the gel pores and transition pores, respectively, and showed a decreasing trend with the increase in GO content. This indicates that GO can reduce the pore size of the corresponding distribution regions to a certain extent. The average pore diameters of HS0, HS1, HS3, and HS5 were 55.9, 46.6, 30.2, and 22.3 nm, respectively. The changes in pores suggested that GO not only filled pores due to its large surface area, but also refined pores due to accelerated hydration reactions. Wang et al. demonstrated that the addition of GO to cement-based materials can refine macro- or capillary pores into transition or gel pores, thus improving the compactness of the interfacial transition layer and inhibiting ion transport [53].

When the specimen was exposed to a dry environment, the water from pores larger than 50 nm evaporated first, but this portion of water hardly changed the concrete’s volume. As the drying time continued, the water from pores smaller than 50 nm (gel pores and partial transition pores) was subsequently lost. The loss of water in the transition pores inevitably led to a loss of balance between the paste and the remaining water, which was the beginning of drying shrinkage. The continuous loss of water in the gel pores caused further drying shrinkage and even cracking [43]. The behavior of GO in refining pores can fundamentally contribute to increasing the drying shrinkage of HSLWC.

### 3.4. Microstructure

Previous studies confirmed that GO can adjust the crystal morphology in cement mortar and UHPC [19,54]. There were intrinsic relationships among drying shrinkage, compressive strength, and the microstructure of HSLWC in this study. In order to systematically analyze the mechanism of drying shrinkage, it was necessary to observe the microstructure of HSLWC. Figure 15 displays the SEM images of random samples made from HS0, HS1, HS3, and HS5 after curing for 28 days. A large number of layered crystals and sheet-like crystals were clearly observed from the sample without GO (HS0), as shown in Figure 15a. In addition, there were nanoscale pores between these crystals. These crystal products were mainly compounds of AFt, C-S-H, CH, and AFm, which also existed in the form of needle-like crystals and rod-like crystals. The shape and quantity of crystals was randomly synthesized in the hydration reaction, which determined the performance of concrete [55].

When the content of GO was 0.01%, some disordered rod-like crystals gradually grew on the fracture surface and pores of the specimen, as shown in Figure 15b. These crystals not only filled in the pores but also formed dense structures. As GO content increased to 0.03%, the growing rod-like crystals developed into clusters of flower-like crystals (Figure 15c). When GO content reached 0.05%, many rod-like crystals grew stronger and finally formed thick and mature flower-like crystals (Figure 15d). During the process of crystal recombination, the volume of HSLWC decreased due to the densification of flower-like crystals. Lv et al. demonstrated that these flower-like crystals have excellent strengthening and toughening abilities, which can improve the mechanical properties and durability of concrete [16].

The experimental results indicated that GO can recombine crystal shapes to form dense flower-like crystals. It also indicated that increasing the content of GO at a lower level was beneficial to the production of more crystals, which can reduce porosity and improve mechanical properties.

## 4. Conclusions

In this study, grade 60 HSLWC was made with SC and SPS as coarse and fine aggregates, respectively. The slumps and 28-day compressive strengths of HSLWC incorporating 0.01%, 0.03%, and 0.05% GO were measured and their drying shrinkage values were investigated for up to 360 days. In order to obtain more valuable support, some typical prediction models were compared with experimental drying shrinkage values, and a modified ACI209 model was developed to accurately predict the drying shrinkage of HSLWC incorporating GO. Furthermore, the mechanism of drying shrinkage was explained in terms of pore structure and microstructure. The main conclusions drawn are as follows:The slump of HSLWC decreased with the increase in GO, which was still in an acceptable range. The low GO content can increase the specific strength of HSLWC by 18.6%.The addition of GO can increase the ultimate drying shrinkage of HSLWC by 1.2%~8.6%. Based on a comparison of typical models, a drying shrinkage prediction model with high accuracy and simple calculation was established, which was derived by modifying the ACI 209 model with the GO content factor. Numerical simulation indicates that the average relative percentage of errors is less than 5%.Pore structure tests indicate that GO can reduce total porosity, which improves the compactness of HSLWC. However, GO can also refine pores, leading to an increase in transition pores and gel pores, which results in the increased drying shrinkage of HSLWC.Microscopic tests indicate that GO can recombine crystal shapes to form dense flower-like crystals, which is helpful in improving the mechanical properties and reducing the volume of HSLWC.In general, GO can significantly improve mechanical properties but is not conducive to drying shrinkage. The modified prediction model can be used to quantitatively grasp the development trend of drying shrinkage in HSLWC so as to investigate the potential hazards in engineering construction. In future studies, other additives can be tried in combination with GO to compensate for drying shrinkage.

## Figures and Tables

**Figure 1 nanomaterials-13-01405-f001:**
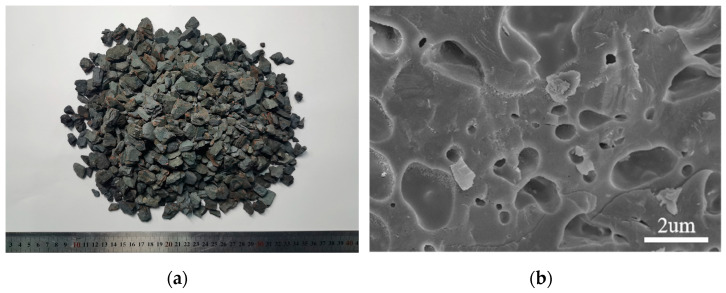
Characteristics of SC: (**a**) physical image; (**b**) microscopic image.

**Figure 2 nanomaterials-13-01405-f002:**
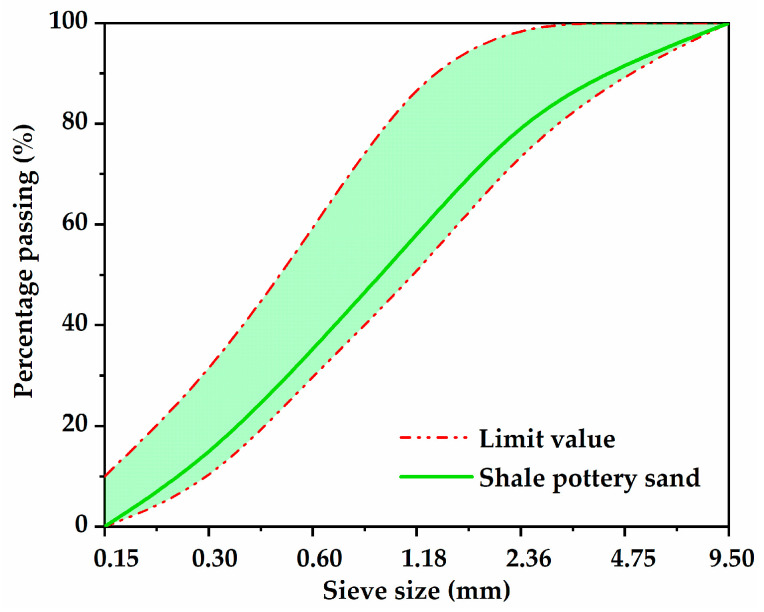
Cumulative particle size distributions of SPS.

**Figure 3 nanomaterials-13-01405-f003:**
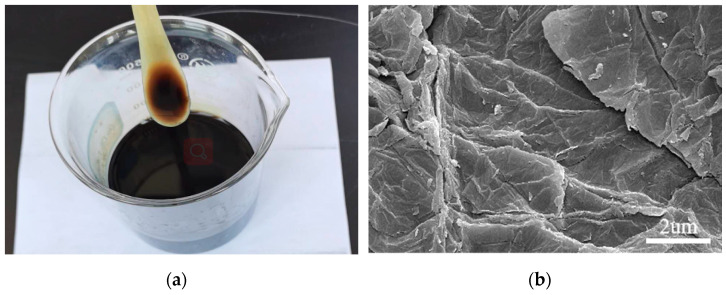
Characteristics of GO: (**a**) GO dispersion; (**b**) microscopic image.

**Figure 4 nanomaterials-13-01405-f004:**
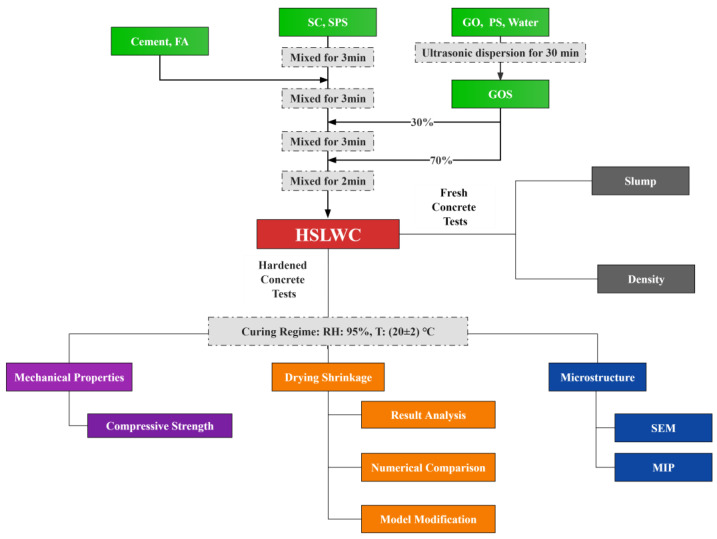
Mixing procedure and experimental items.

**Figure 5 nanomaterials-13-01405-f005:**
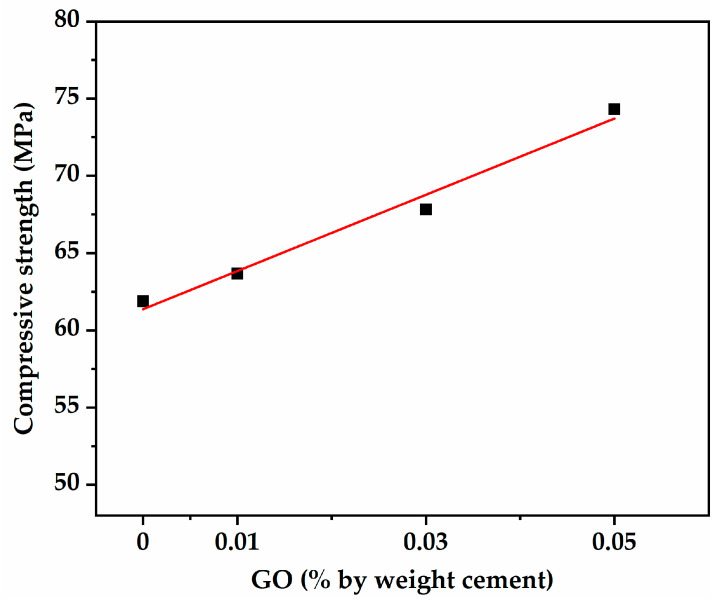
The relationship between GO content and compressive strength.

**Figure 6 nanomaterials-13-01405-f006:**
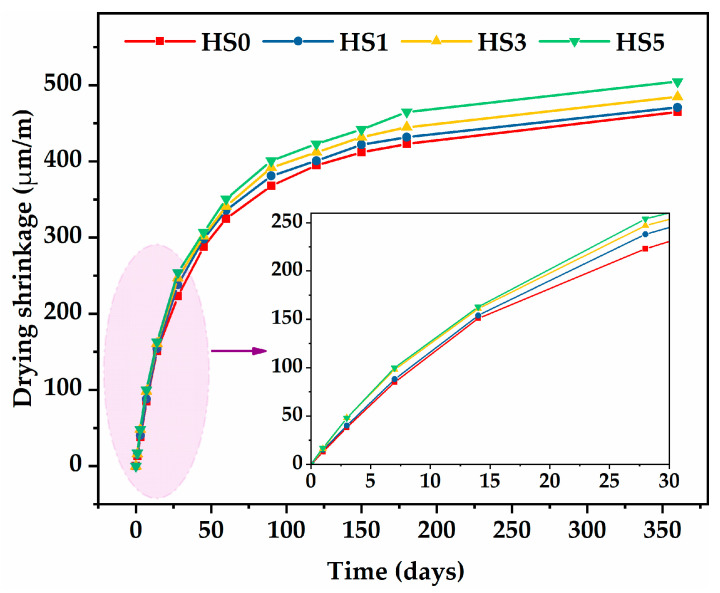
Drying shrinkage developments of HSLWC with different GO content.

**Figure 7 nanomaterials-13-01405-f007:**
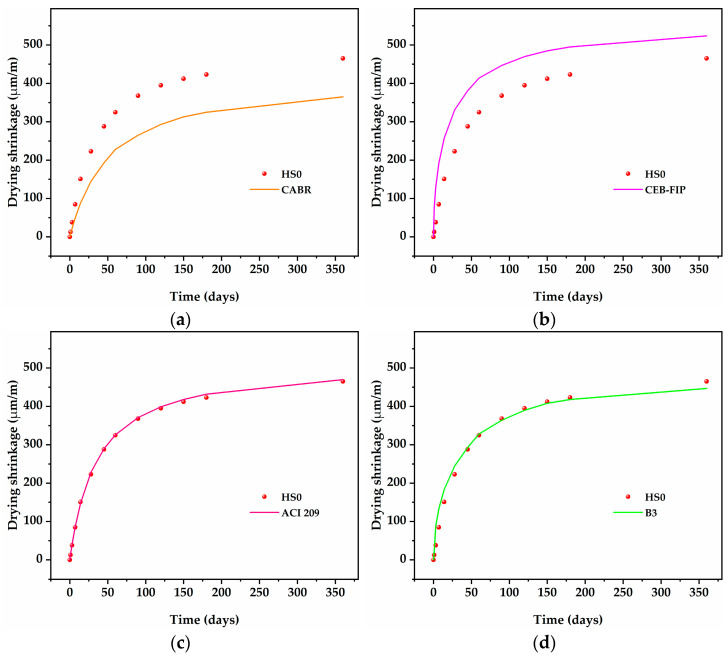
Comparisons of the prediction models and experimental shrinkage values of HSLWC: (**a**) CABR; (**b**) CEB-FIP; (**c**) ACI 209; (**d**) B3; (**e**) GL2000; (**f**) SAKATA.

**Figure 8 nanomaterials-13-01405-f008:**
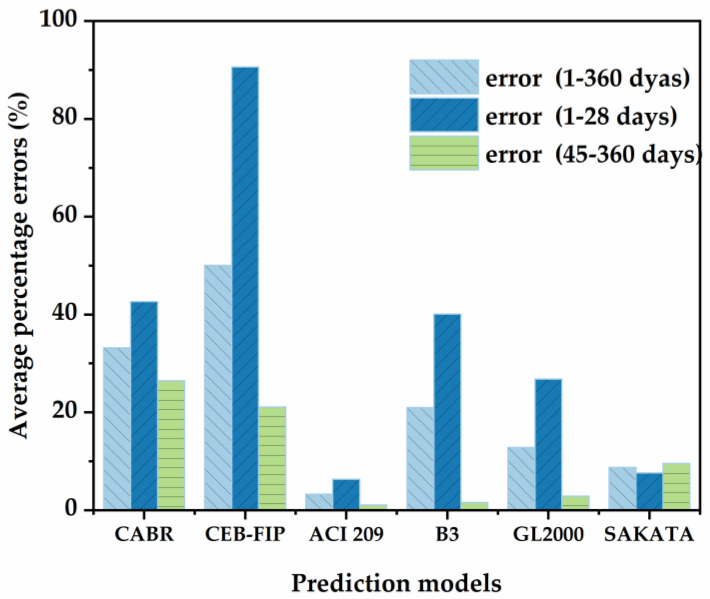
Average percentage of errors for the various prediction models.

**Figure 9 nanomaterials-13-01405-f009:**
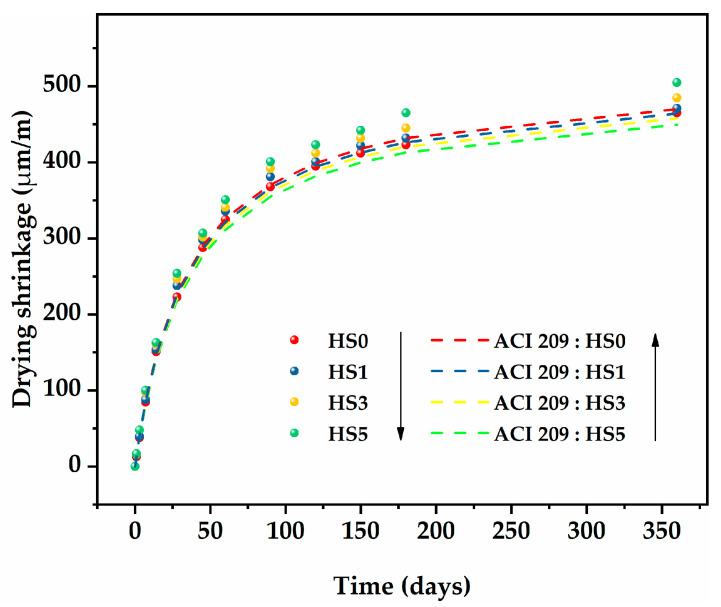
Comparison of the ACI 209 model and experimental shrinkage values of HSLWC with different GO content.

**Figure 10 nanomaterials-13-01405-f010:**
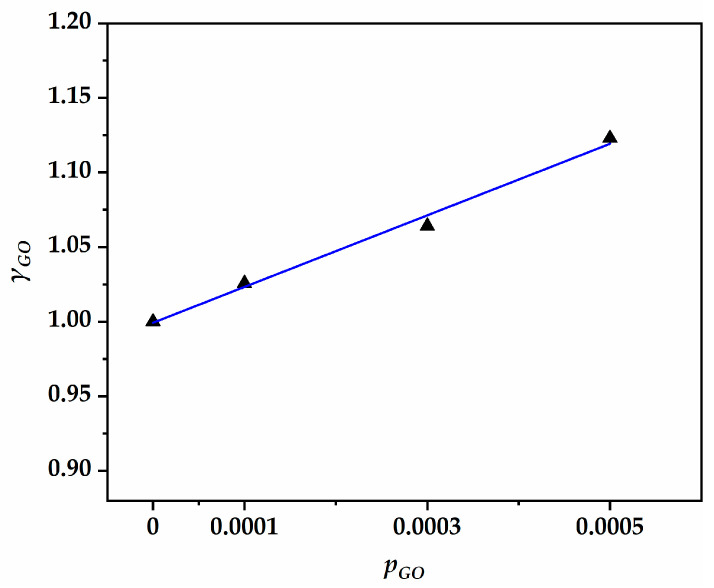
The fitting relationship between $p_{GO}$ and $\gamma_{GO}$.

**Figure 11 nanomaterials-13-01405-f011:**
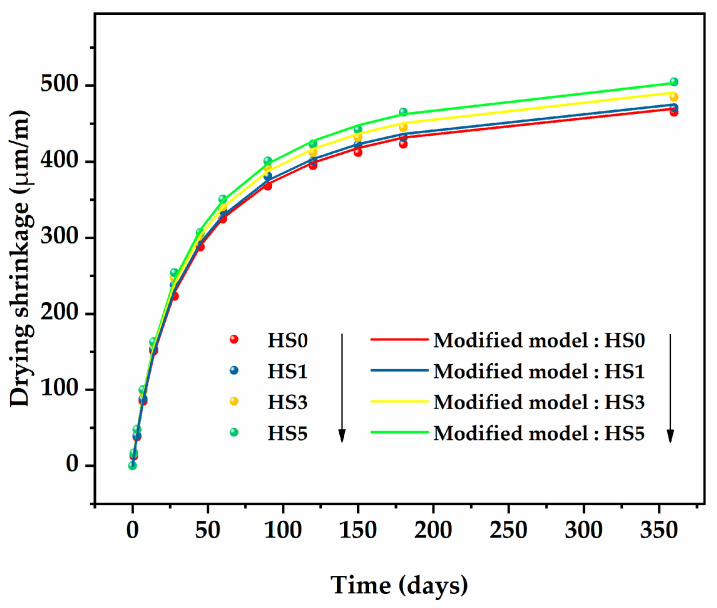
Comparison of the modified ACI 209 model and experimental shrinkage values of HSLWC with different GO content.

**Figure 12 nanomaterials-13-01405-f012:**
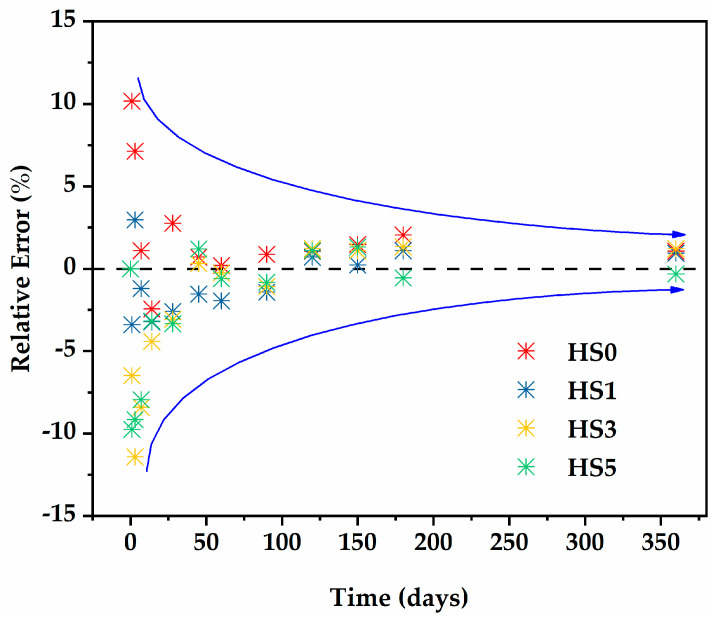
Relative errors of the predictions with different GO content.

**Figure 13 nanomaterials-13-01405-f013:**
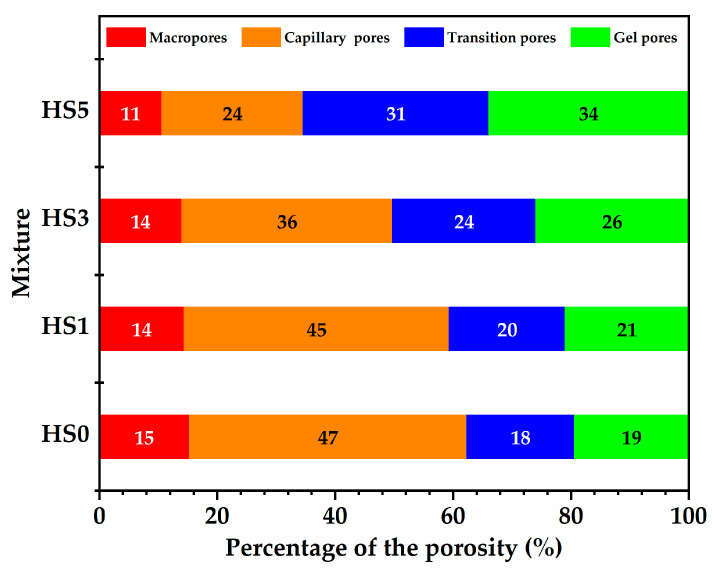
Percentages of pores of different pore size for each mixture.

**Figure 14 nanomaterials-13-01405-f014:**
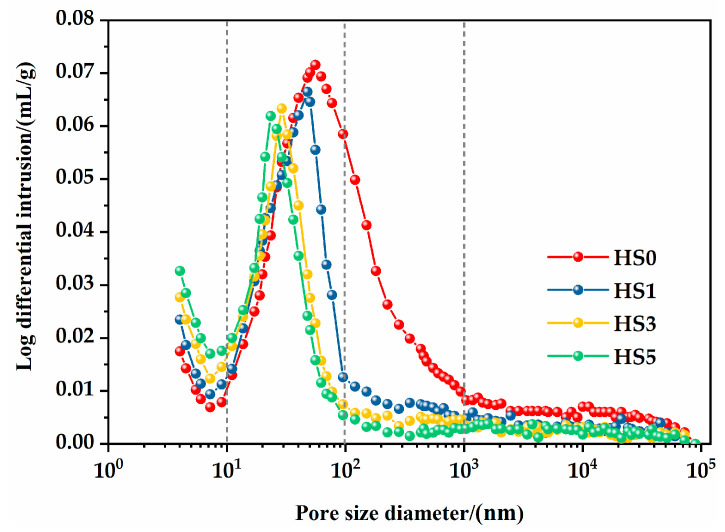
The pore size distributions of individual mixtures.

**Figure 15 nanomaterials-13-01405-f015:**
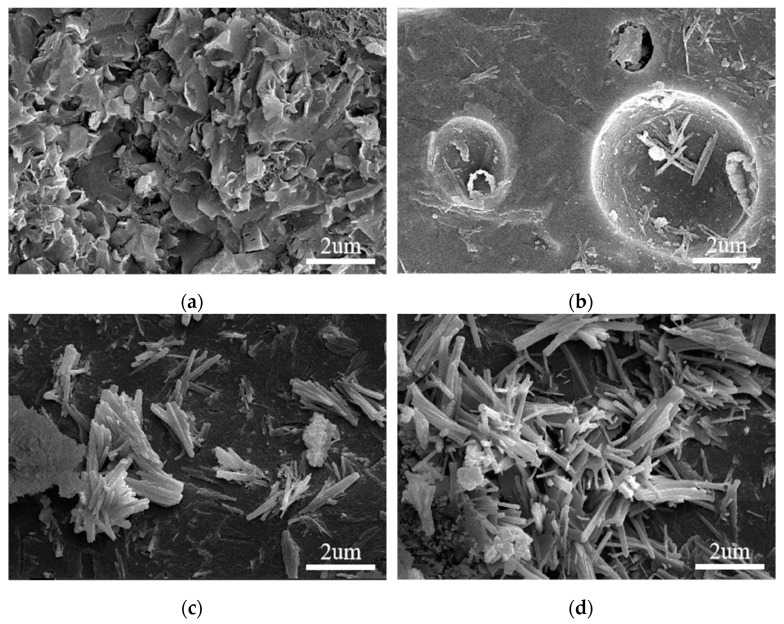
SEM images of different mixtures at day 28: (**a**) HS0; (**b**) HS1; (**c**) HS3; (**d**) HS5.

**Table 1 nanomaterials-13-01405-t001:** Physical properties of SPS and SC.

Aggregate	Density Rank (kg/m^3^)	Diameter (mm)	Apparent Density (kg/m^3^)	Water Absorption (3 h) (%)	Water Absorption (24 h) (%)	Cylinder Compressive Strength (MPa)
SPS	700	<5	1638	1.2	1.4	-
SC	800	5–15	1425	2.9	4.6	5.2

**Table 2 nanomaterials-13-01405-t002:** Properties of GO.

Oxygen Content (%)	Carbon Content (%)	Purity (%)	Thickness (nm)	Diameter (µm)	Specific Surface Area (m^2^/g)
>33	>66	>95	~1	10–30	100–300

**Table 3 nanomaterials-13-01405-t003:** Mix proportions (kg/m^3^).

Mix No.	Cement	FA	SC	SPS	Water	SP	GO
HS0	440	110	380	380	170	11	0
HS1	440	110	380	380	170	11	0.044
HS3	440	110	380	380	170	11	0.132
HS5	440	110	380	380	170	11	0.220

**Table 4 nanomaterials-13-01405-t004:** Slumps, densities, and compressive strengths of HSLWC.

Mix No.	HS0	HS1	HS3	HS5
Slump (mm)	113	105	96	84
Oven Dry Density (kg/m^3^)	1696	1701	1705	1715
Compressive Strength (MPa)	61.88	63.67	67.81	74.32
Specific Strength (kN·m/kg)	36.5	37.4	40.3	43.3

**Table 5 nanomaterials-13-01405-t005:** Typical prediction models.

Model	Year	Ref.	Equation
CABR	1986	[44]	$\varepsilon_{sh(t)}={\varepsilon (t)}_{0}\times \beta_{1}\times \beta_{2}\times \beta_{3}\times \beta_{5}\times \beta_{6}$
CEB-FIP	1990	[45]	$\varepsilon_{sh(t)}=\varepsilon_{shu}\beta \left(h\right)\beta \left(t\right)$
ACI 209	1992	[46]	$\varepsilon_{sh(t)}=\frac{t}{35+t}\varepsilon_{sh,\infty }$
B3	1996	[47]	$\varepsilon_{sh(t,t_{0})}=-\varepsilon_{sh\infty }k_{h}S(t)$
GL2000	2000	[48]	$\varepsilon_{sh(t)}=\varepsilon_{shu}\beta \left(h\right)\beta \left(t\right)$
SAKATA	2001	[49]	$\varepsilon_{sh(t,t_{0})}=\frac{\varepsilon_{sh\infty }.\left(t-t_{0}\right)}{\beta +\left(t-t_{0}\right)}$

**Table 6 nanomaterials-13-01405-t006:** Factors considered for shrinkage prediction models.

NO.	Factors	Prediction Models					
		CABR	CEB-FIP	ACI 209	B3	GL2000	SAKATA
(1)	Fine Aggregate Content			●			
(2)	Water Content		●		●	●	●
(3)	Cement Content			●			
(4)	Cement Type			●	●	●	●
(5)	Air Content			●			
(6)	Slump			●			
(7)	Fly Ash Replacement Ratio	●					
(8)	Compressive Strength	●	●		●	●	●
(9)	Maintenance Method	●		●	●		
(10)	Relative Humidity	●	●	●	●	●	●
(11)	Component Section Size	●	●	●	●	●	●

## Data Availability

Data are contained within the article.

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
