# Peer review of "Prediction Model and Mechanism for Drying Shrinkage of High-Strength Lightweight Concrete with Graphene Oxide"

_nanomaterials, 2023, doi:10.3390/nano13081405_

Round 1

Reviewer 1 Report

In this study, the authors investigate the compressive strength and drying shrinkage behavior of HSLWC incorporating low GO content (0.00%-0.05%), focusing on the prediction and mechanism of drying shrinkage. Overall speaking, this is an interesting study. The experiments are well designed and the conclusions are well supported by the results. Below are my comments:

1. Drying shrinkage is bad for concrete. In the abstract, the expression like "Drying shrinkage can be maximized by 8.6%" sounds like you want to see a high drying shrinkage. Similarly, "which contributes to the drying shrinkage". Please also check the other part of the paper.

2. The introduction should also focus on the mechanism of drying shrinkage. The loss of water in capillary pores causes the shrinkage stress. The recently published papers should be reviewed. (e.g., "Experimental and thermodynamic study of alkali-activated waste glass and calcium sulfoaluminate cement blends: shrinkage, efflorescence potential, and phase assemblages. Journal of Materials in Civil Engineering33(11), p.04021312.")

3. The dispersion of GO should be described in detail. What is used to disperse GO? How much GO exists in the dispersion liquid?

4. What are the flower like crystals in SEM?

5. What is transition pore? Do you mean the pore connectivity is increased?

Reviewer 2 Report

This paper presented the experimental research on the prediction models and mechanisms on the mechanical properties and drying shrinkages of high-strength lightweight concrete (HSLWC) incorporating varying contents of the nanomaterial graphene oxide. The obtained experimental results are genuine, initiative, extensive and practically useful, and the discussion and conclusions are comprehensive, sound and convincing. The paper was fairly well prepared, including tables, figures and references. The paper provides much new and useful information so it is worthwhile to publish after review. However, there are numerous technical, editorial and grammatical errors and they need to be corrected before the paper can be finally published. In References, all the journal paper titles should be in lower case except the first letter. I have marked them in the PDF file submitted by the authors. The authors should pay attention to each of these and mark the revisions clearly in a different colour in the revised manuscript for re-review.

Author Response

Please see the attachmen!

Reviewer 3 Report

The article is related to the development of high-strength lightweight concrete with graphene oxide. Especially focusing on the development of the prediction and mechanism on drying shrinkage of such kind of concrete is very interesting for engineering and scientific community. The manuscript is well written. I have just some minor comments before further processing:

- I suggest to change the word “incorporating” to “with” in the title,

- please do not use abbreviations as keywords (e.g. GO; HSLWC) but rather use full terms,

- fig. 4 is not necessary since it does not add much to the content of the article. I suggest to delete it,

- fig. 5 and 10 – four points are not enough to calculate any correlation – I suggest to delete R2 or provide a proof that this correlation is proper from the scientific point of view,

- I would like to see some perspectives in conclusions section.

Reviewer 4 Report

nanomaterials-2311536

Title: "Prediction model and mechanism on drying shrinkage of high-strength lightweight concrete incorporating graphene oxide"

The authors did a good job of studying the Prediction model and mechanism on drying shrinkage of high-strength lightweight concrete incorporating graphene oxide; This scope needs such studies. However, in order to raise the quality of the manuscript to bring it to the required level in the journal “Nanomaterials”, the reviewer recommends a deep processing of the comments (attached).

Round 2

Reviewer 1 Report

This paper has been revised based on the comments and should be ready for publication.

Reviewer 4 Report

Manuscript well revised and can process for next stage of publication.